# Blood-Spinal Cord Barrier: Its Role in Spinal Disorders and Emerging Therapeutic Strategies

Neha Chopra [1,2], Spiro Menounos [1] , Jaesung P. Choi [3] , Philip M. Hansbro [3] , Ashish D. Diwan [1,2] and Abhirup Das [1,2,*]

1    Spine Labs, St. George & Sutherland Clinical School, University of New South Wales,
     Kogarah, NSW 2217, Australia; Neha@spine-service.org (N.C.); s.menounos@student.unsw.edu.au (S.M.);
     A.Diwan@spine-service.org (A.D.D.)
2    Spine Service, St. George Hospital, Kogarah, NSW 2217, Australia
3    Centre for Inflammation, Faculty of Science, Centenary Institute, School of Life Sciences, University of
     Technology Sydney, Sydney, NSW 2050, Australia; Jaesung.Choi@uts.edu.au (J.P.C.);
     Philip.Hansbro@uts.edu.au (P.M.H.)
*    Correspondence: abhirupdas@unsw.edu.au

**Abstract:** The blood-spinal cord barrier (BSCB) has been long thought of as a functional equivalent to the blood-brain barrier (BBB), restricting blood flow into the spinal cord. The spinal cord is supported by various disc tissues that provide agility and has different local immune responses compared to the brain. Though physiologically, structural components of the BSCB and BBB share many similarities, the clinical landscape significantly differs. Thus, it is crucial to understand the composition of BSCB and also to establish the cause–effect relationship with aberrations and spinal cord dysfunctions. Here, we provide a descriptive analysis of the anatomy, current techniques to assess the impairment of BSCB, associated risk factors and impact of spinal disorders such as spinal cord injury (SCI), amyotrophic lateral sclerosis (ALS), peripheral nerve injury (PNI), ischemia reperfusion injury (IRI), degenerative cervical myelopathy (DCM), multiple sclerosis (MS), spinal cavernous malformations (SCM) and cancer on BSCB dysfunction. Along with diagnostic and mechanistic analyses, we also provide an up-to-date account of available therapeutic options for BSCB repair. We emphasize the need to address BSCB as an individual entity and direct future research towards it.

**Keywords:** blood-spinal cord barrier (BSCB); spinal cord injury (SCI); amyotrophic lateral sclerosis (ALS); degenerative cervical myelopathy (DCM); peripheral nerve injury (PNI); ischemia reperfusion injury (IRI); multiple sclerosis (MS); spinal cavernous malformations (SCM)





## 1. Introduction

Blood vessels are essential for delivering oxygen and nutrients throughout the body. In the vascular tree, the controlled communication that occurs between blood vessels and components of central nervous system (CNS) is unique. Physiologically, there are three specialised interfaces in the human body that selectively permit entry of nutrients, ions, lipids and small molecules from the blood stream to either the brain (blood-brain barrier; BBB), cerebral spinal fluid (blood-cerebral spinal fluid barrier; BCSFB) or spinal cord (blood-spinal cord barrier; BSCB). Of these the BBB is the most studied and its dysfunction is associated with neurological disorders such as multiple sclerosis (MS), Alzheimer's disease. and Parkinson's disease [1]. Recent evidence suggests that BBB dysfunction is an underlying mechanism associated with age-related neuronal deterioration [2]. As a result of improved understanding of the morphology and the consequences of dysfunction of the BBB, various translational drugs and models have been realised.

Drugs such as Natalizumab, a humanized monoclonal antibody acting on the VLA-4/VCAM-1 axis, modulates BBB leakage and inhibits the entry of T cells into the CNS in relapsing MS patients [3]. In Alzheimer's disease, "Trojan horse" strategies have been used,

wherein a two-sided antibody crosses the BBB and degrades β-secretase (a precursor to amyloid β protein) while remaining bound to transferrin receptor with one arm [4]. Clinical trials using strategies such as ultrasound waves to deliver drugs across the BBB are being tested for glioblastoma patients [5]. More recently, in vitro "organoids" representing BBB dysfunction have been developed as a potential platform for drug testing and development of therapeutics [6].

Although the role of BBB is well studied in brain disorders, its functional equivalent, the BSCB, lacks in-depth investigation regarding its role in spinal cord and neurological disorders as it is still considered as simply an extension of the BBB. Here, it is notable that although the constituents of both the BBB and BSCB are similar, yet functional differences in their role in diseases of the spinal cord differ [7]. Compared to the brain, the spinal cord is a far more agile organ that is surrounded by supporting diverse tissues and extracellular matrices. In the event of dysfunction, immune responses generated from the spinal cord clinically differ from those in the brain and require a suitable intervention. Dysfunction of the BSCB is associated with various traumatic and non-traumatic injuries, aging and cancer and should be considered as a separate entity for research and clinical purposes. The vascular structure of the BSCB, imaging techniques and role in diseases have been elegantly summarised by Bartanusz et al. [7]. However, over the last decade, further scientific revelations provide further detailed insight of the role of BSCB in various neurodegenerative diseases and warrants an update. Here, we fill the existing gap in the literature pertaining to the role the BSCB and BSBC in maintaining spinal cord health and how its dysfunction leads to different disorders.

## 2. Anatomy of BSCB

The structural components of BSCB are (see Figure 1):

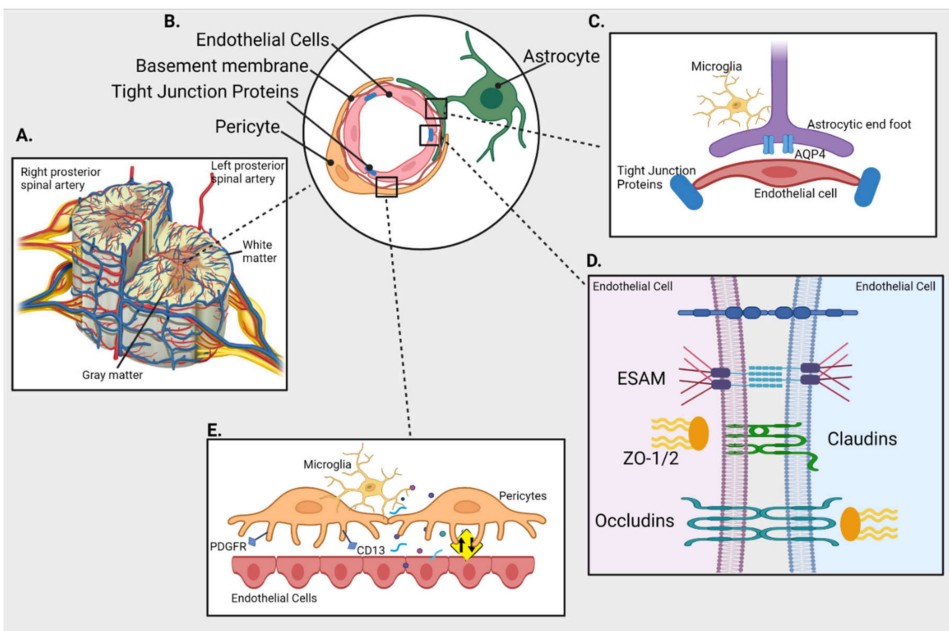

**Figure 1.** Blood-spinal cord barrier (BSCB) and its cellular components. (**A**) Cross section of spinal cord and its blood supply (**B**) Schematics of cellular components of BSCB; endothelial cells (ECs) are separated from pericytes by basement membrane and held together by tight junction proteins (TJs); astrocytes connect with both pericytes and ECs (**C**) Inset shows interaction between the astrocytic end foot enriched with aquaporin 4 (AQP4) and ECs in presence of immune cells (microglia) (**D**) Inset shows TJs like endothelial cell-selective adhesion molecule (ESAM), claudins, occludins and Zonula occludens (ZO-1) holding the ECs together (**E**) Inset represents interaction between ECs and pericytes (expressing PDGFR and CD13) that plays an important role in maintaining the integrity of BSCB by regulating the uptake of circulating macromolecules (presented as yellow sign).

### 2.1. Spinal Cord Microvascular Endothelial Cells (SCMECs)

These are the outermost cellular structures that form the first point of contact for any incoming molecule and restrict or allow entry of molecules between the blood and spinal cord. SCMECs act like molecular gateways and restrict the trans-flow of molecules from the blood stream to the spinal cord [8]. Endothelial cells are sealed by tight junctions (TJs) and have limited endocytic vessels. The presence of more glycogen deposits in the spinal cord than the cerebral vasculature could be related to high metabolic activity due to increased number of mitochondria in SCMECs [7].

### 2.2. Basal Membrane

The basal membrane or basal lamina comprise mainly of laminin, collagens, fibronectins, proteoglycans that provide structural integrity to the abluminal surface of SCMECs. It also engulfs and separates pericytes from endothelial cells and plays a role in blocking the entry of macromolecules [9].

### 2.3. Pericytes

Pericytes are small vessel-walled cells which are crucial for the maintenance of the BSCB. They influence the proliferation, migration, and differentiation of endothelial cells by forming the TJs and flow of soluble factors. Furthermore, pericytes are also associated with the expression and alignment of TJ proteins and reduced uptake of circulating macromolecules thereby maintaining the integrity of the BSCB [1,10,11]. Pericytes are a heterogenous cell population and a universal marker for their identification has not been defined. The most commonly used markers are aminopeptidase-N (CD13) and platelet-derived growth factor receptor (PDGFR) [8].

### 2.4. Astrocytic Feet

Adjoining the basal lamina, are astrocytic foot processes that maintain the functional and structural integrity of the barrier characteristics such as polarity, permeability and even re-vascularization [9,12]. These spinal-cord-astrocytes end foot processes also impart neuroprotective mechanisms to SCMECs and also highly express aquaporin-4 and potassium channels Kir4.1 that regulate resting potassium ion conductance and fluid volume in the spinal cord [7].

### 2.5. TJ Proteins

TJs are diffusion barriers situated between endothelial cells. TJ proteins can be categorised into (i) claudins (~27 member proteins); (ii) TJ-associated MARVEL proteins (TAMPs; MARVEL motif, occluding, tricellulin, MarvelD3); (iii) Immunoglobulin superfamily membrane proteins (JAM-A/-B/-C, coxsackie adenovirus receptor, endothelial cell-selective adhesion molecule (ESAM)) [13]. Of these, claudin-1, claudin-5 and occludin, as well as Zonula occludens (ZO-1) primarily constitute the BSCB [8]. The relative higher permeability of the BSCB compared to the BBB has been attributed to the reduced levels of specific TJ proteins such as occludin and ZO-1 [7]. Furthermore, disruptions in the expression of these TJ proteins are associated with BSCB leakage and the onset of various neurological disorders [14]. Regulatory pathways such as GSK3β are regulators of TJ protein transcription and translation via transcription factor enhancers such as catenin or suppressors of the snail family of zinc finger transcription factors [8].

### 2.6. Transporters

Though further research is required to identify key transporters; a few studies have identified high expression of the ABC efflux transporter-ABCA2 in the spinal cord [15,16]. More recently, Uchida et al., compared the abundance of transporter proteins in different regions of the brain and showed expression of identical proteins within a twofold range in the BBB and BSCB whilst many other proteins had >twofold lower expression in the BSCB [17]. Their key findings suggest higher expression of receptors such as insulin

receptor (INSR), low-density lipoprotein receptor-related protein 1 (LRP1) and GLUT1 at corticol BBB than BBB and BSCB white matter. Aquaporins (AQPs) are important membrane proteins that regulate water movement through the BSCB. Of these, AQP-1 and -4 are key transporters that mediate water absorption in hypo-osmotic conditions [18].

For a better understanding, the morphological and physiological differences between BBB and BSCB have been highlighted in Table 1.

**Table 1.** Key differences between BBB and BSCB.

| Feature | Blood-Brain Barrier (BBB) | Blood-Spinal Cord Barrier (BSCB) | References |
|---|---|---|---|
| Permeability | Low | High: 3H-mannitol and $^{14}$C-inulin | [19] |
| Tight Junction proteins | High | Low: ZO-1, occludin, β-catenin, VE-cadherin | [20] |
| Number of pericytes | High | Low | [21] |
| Glycogen Deposits | Low | High | [7] |

### 3. Methods to Assess BSCB Impairment

The BSCB is relatively more permeable compared to the BBB [22]. This can be attributed to differences in the composition of TJ proteins that allows secretory factors to transverse through the interface. This phenomenon was initially evidenced by Prockop et al., wherein elegant experiments showed that the kinetic transfer of [$^3$H]-*D*-mannitol and [$^{14}$C]-carboxyl-inulin was higher in regions of the spinal cord compared to the brain in rabbits [19]. Further, permeability of various other cytokines such as IFN-$\alpha/\gamma$ and TNF-$\alpha$ was higher in spinal cord compared to brain regions [23]. In order to assess the BSCB dysfunction, various tracer dyes and imaging techniques have been used.

#### 3.1. Dye Extravasation

Evans blue dye is one of the most extensively used markers/tracers to detect BSCB leakage owing to its strong affinity towards serum albumin minimizing false-negative staining [24]. Extravasation of Evans blue has been used as evidence of BSCB disruption in models of experimental autoimmune encephalomyelitis (EAE) and spinal cord injury (SCI) [25,26]. However, additional evidence shows that Evans blue can bind to other proteins such as globulins, $\alpha$1-lipoprotein, hemopexin and transferrin raising concerns over detection of barrier leaks [27]. Furthermore, toxicity concerns of Evans blue are also a major limitation for in vivo studies. Over the years, less toxic dyes such as sodium fluorescein (Na-F) have shown greater sensitivity in predicting BBB/BSCB leakage in amyotrophic lateral sclerosis (ALS) models and superiority over fluorescent tracers like fluorescein isothiocyanate (FITC)-labelled albumin and FITC-labelled dextrans-70 [28]. As well as dyes, estimating immunoglobulins (Ig) has also been used as a measure of BSCB dysfunction in degenerative cervical myelopathy (DCM) [29] and ALS [30].

#### 3.2. Contrast Magnetic Resonance imaging (MRI)

In vivo imaging of the spinal cord is technically challenging given its small size and related motion artifacts associated with cardiovascular and respiratory organs [31]. This is a major reason for limited studies in this field thereby impacting clinical translation. In this respect, a recent systematic review by Bakhsheshian et al. evaluated various in vivo imaging techniques used to assess different diseases. They found that MRI and intravital microscopy (IVM), are the most common techniques used to assess the BSCB in rodent models of EAE and SCI [32]. Dynamic contrast-enhanced MRI (DCE-MRI) is another sensitive, non-invasive technique that can be used to assess BSCB permeability in mouse models with peripheral nerve injury (PNI) [33].

#### 3.3. Immunohistochemistry (IHC)

The cause and effect of BSCB leakage in SCI has been assessed with infiltration of immune cells and activation of certain pathways (e.g., shh/Gli) using techniques such as

IHC [34,35]. Using IHC, a direct correlation between expression of vimentin in spinal endothelium and astrocytes and pain has been established in nerve injuries [36]. A variety of other molecules such as Amigo2, aquaporin 4, CD3, CD34, GFAP, ionized calcium-binding adapter molecule 1 (Iba1), myelin basic protein (MBP), non-phosphorylated neurofilaments (np-NF), periaxin, S100A10, CCL2 and TMEV have been evaluated using IHC in cervical and thoracic spinal cord segments in Theiler's murine EAE [37,38]. Vascular disturbances in ALS were captured by IHC where microvascular density (MVD) and pericyte coverage (PC) were quantitatively evaluated to understand their role in BSCB impairment [35].

### 3.4. Electron Microscopy

Electron microscopy has been used to validate polarization of AQP4 in perivascular astrocytes in Theiler's murine EAE. This technique could also provide insights into the irregular vasculature of astrocytes that contribute to disease severity [37]. Similarly, in ALS, the frequency of degenerated endothelium and pericytes, vacuolar changes in endothelial cytoplasm in these cells were evaluated in ALS patients [39]. More recently, Ying et al., used transmission electron microscopy (TEM) to evaluate the role of BDNF/TrkB-CREB signalling pathway in tread mill training mediated BSCB protection after SCI [40]. The impairment of BSCB has been implicated in the pathophysiology of various spinal-cord related disorders such as SCI and ischemia, ALS, PNI, DCM and MS. We discuss how BSCB impairment contributes to the pathogenesis of these disorders and diseases.

## 4. Spinal Cord Disorders
### 4.1. SCI

Activation of the matrix metalloprotease (MMPs) has been regarded as a triggering event for BSCB dysfunction. Under normal physiological conditions, MMPs play a role in processes such as tissue morphogenesis, angiogenesis, cell migration, wound healing and inflammation. However, their role also encompasses degradation of various components of the extracellular matrix (ECM) thereby permitting infiltration of cells, leukocytes, etc. to breach barriers such as BBB and BSCB [41,42]. MMP-3 and -9 are regulated via upregulation of histone H3K27 demethylase Jmjd3 leading to loss of TJ proteins and increase in BSCB permeability [43,44]. Similarly, others such as MMP-8 and -12 have also shown alterations in the abundance of TJ protein and barrier permeability as an after effect of SCI [45,46]. The mechanical stress due to SCI can damage cellular components of BSCB such as the endothelial cells via enhanced expression of cation channels such as transient receptor potential vanilloid type 4 (TRPV4) [47]. Downregulation of USP-4 after SCI promotes microglial activation and neuronal inflammation (TNF-α and IL-1β) via NF-κB by attenuating the de-ubiquitination of TRAF629. These activated microglia/macrophages can modulate neural regeneration based on their polarization (M1/M2) [48]. This phenomenon has been observed in aldose reductase (AR) knock-out mice where smaller lesion areas were observed post-SCI due to induction of M2 response as compared to the wild-type group (where AR is upregulated). The authors have implied that AR can work as a switch to regulate microglia polarization to either M1 or M2 phenotype through cAMP Response Element-Binding Protein30. Apart from NF-κB signalling, Shh/Gli is another signalling pathway which is induced in reactive astrocytes post SCI [49]. A cumulative analysis of the mechanisms and key cells of the BSCB affected post SCI have been outlined in Table 2.

**Table 2.** Implications of blood-spinal cord barrier dysfunction in different spinal cord disorders.

| Cells | Cause/Effects of BSCB Impairment | Refs. |
|---|---|---|
| | **Spinal Cord Injury (SCI)** | |
| Microglia | Jmjd3 ↑ → NF-κB → MMP3 ↑ and MMP9 ↑ | [44] |
| | TRPV4 ↑ → spinal cord scarring, endothelial damage, BSCB damage | [47] |
| | MMP-8 ↑ → occludin ↓ and ZO-1 ↓ | [45] |
| | MMP-12 → functional recovery↓, BSCB permeability ↓ | [46] |
| | USP4 ↓; NF-κB → TRFAF6 → Neuronal inflammation | [48] |
| | AR deficiency → M2 response → locomotion recovery<br>AR inhibition → HNE accumulation → phosphorylation of CREB → Argl ↑ | [49] |
| | AQP4 ↑ → BSCB permeability ↑ | [50] |
| Reactive astrocytes (RAs) | Shh/Gli ↑ → BSCB permeability ↑, locomotor recovery ↓ | [25] |
| | Ras → fibronectin/β1 integrin pathway → microglial inflammation | [51] |
| | Calmodulin → AQP4 ↑ | [52] |
| Macrophages | Perforin ↑ → BSCB permeability ↑ → cytokine and inflammatory cell infiltration | [49,51,53] |
| Neutrophils | MMP-3 ↑ → NF-κB → occludin ↓ and ZO-1 ↓ | [43] |
| | **Amyotrophic Lateral Sclerosis (ALS)** | |
| Astrocytes | Wnt7a ↓, Wnt5a ↓<br>Gi signalling in astrocytes restores BSCB integrity | [54] |
| | Swollen astrocyte foot processes<br>Degenerating astrocytes | [55] |
| | Glutamate ↑ → EC P-gp ↑, NMDA ↑ | [56] |
| Neurons | Motor neuron loss | [9,30,35,54–63] |
| | PDGFC ↑ → BSCB dysfunction | [57] |
| Immune cells | Erythrocyte extravasation | [30] |
| Pericyte | Reduction in pericytes | [30,35] |
| Endothelial cells | Glut-1 ↓, CD146 ↓ | [9] |
| | Claudin 5 ↓, occludin ↓, ZO-1↓ | [60,62,63] |
| | Cytoplasmic vacuoles | [61] |
| | Mitochondrial degeneration | [55] |
| | P-gp ↑, BCRP ↑, MRP2 ↑ | [56,57,59] |
| | ROS ↑ | [62] |
| | Circulating ECs ↓ | [58] |
| ECM | Agrin ↓ | [60] |
| | **Peripheral Nerve Injury (PNI)** | |
| Microglia | MCP-1 ↑; EB extravasation ↑; IL-1β ↑; TGF-β1 ↑; ZO-1, occludin ↓ | [64] |
| | MCP-1 → microglial activation → neuropathic pain → delayed astrocyte activation | [65] |
| Astrocytes | AQP4 ↑ → length and volume of astrocytic processes ↑ | [66] |
| | SUR1-TRPM4 ↑ → dorsal horn astrocytes | [67] |
| | Degenerative Cervical Myelopathy (DCM) | |
| Immune cells | Angiopoietin 2 ↓, VEGF C ↓ | [68] |
| | Peripheral monocytes ↑ | [69] |
| | IgGA ↑, IgGQ ↑, BSCB permeability ↑ | [29] |

**Table 2.** *Cont.*

| Cells | Cause/Effects of BSCB Impairment | Refs. |
|---|---|---|
| | **Multiple Sclerosis** | |
| Endothelial cells | Claudin-11 ↓; BSCB permeability ↑ | [70] |
| Microglia | TMP → STAT3/SOC3 → NF-κB → M1 to M2 polarization<br>TNF-α ↑, IL-1β ↑, IL-4 ↓, IL-10 ↓ | [26] |
| Neutrophils | IL-R1 → adhesion of neutrophils to inflamed SC | [71] |
| | **Spinal Cord Ischemia** | |
| Microglia | TLR4/MyD88/TRIF ↓<br>Inflammation ↓ | [72] |
| | HMGB1 ↑ | [73] |
| | CXCL13/CXCR5 ↑ → ERK | [74] |
| | TUG ↓ → TRIL ↓ → NF-κB/IL-1β ↓ | [75] |
| | Nrf2 ↑ → p53/p38/MAPK/NF-κB → ABC transporters | [76] |
| | **Cancer** | |
| Microglia | ZO-1 ↓, claudin-5 ↓ | [77] |

PDGF = Platelet-derived growth factor; MRP2 = Multidrug resistance protein; EC = Endothelial cell; NMDA = N-methyl-D-aspartic acid; BCRP = Breast cancer resistance protein; ZO = Zona occludens; NF-κB = Nuclear factor kappa B; AR = Aldose reductase; HNE = 4-hydroxynonenal; RAs = Reactive astrocytes; TMP = Tetramethylpyrazine; CREB =cAMP response element-binding protein; TUG = Taurine-upregulated gene 1; HMGB1 = high-mobility group box-1; TLR4 = Toll-like receptor 4; TRIL = TLR4 interactor with leucine-rich repeats; MMP = Matrix metalloproteinases; BSCB = Blood-spinal cord barrier; MCP-1 = Monocyte Chemoattractant Protein-1; SC = Spinal cord; TLR4 = Toll-like receptors 4; MyD88 = Myeloid differentiation factor 88; TRIF = TIR domain-containing adaptor inducing IFN-β.

Ample evidence has been provided especially by Sharma et al., where a variety of neuroprotective agents such as neurotrophins, peptide hormones [78], antioxidants [79] and bradykinin antagonist [80] have been shown to attenuate BSCB disruption evidenced by significant reductions in extravasation of protein tracers (Evans blue, iodine or lanthanum tracers). In this respect, attenuation of the SUR1/TrpM4 (known to mediate haemorrhage) and MMP-9 expression using a variety of compounds such as hormones (ghrelin, 17β-estradiol) and others (protocatechuic acid, flufenamic acid) restores the BSCB by modulating the infiltration of neutrophils and macrophages/microglia [81–84] (Table 3). The recruitment of monocyte-derived macrophages, post SCI, is in fact facilitated by brainventricular choroid plexus (ChP), a compartment of the BCSF, indicating its co-ordinated actions with BSCB [85]. Studies have shown that modulation of IFN-γ/IFN-γR expression by ChP could boost recruitment of anti-inflammatory molecules to the site of injury and may be a novel treatment approach [86]. Furthermore, pathways such as STAT1-NF-κB have been modulated using acidic compounds such as valproic, salvianolic and oleanolic acids to reduce pro-inflammatory responses in SCI by restoring BSCB permeability [87–89]. Other chemical/protein compounds with therapeutic potential in SCI induced BSCB disruption have are listed in Table 3.

**Table 3.** Therapeutic options for blood-spinal cord barrier in different spinal cord disorders.

| | Intervention | Mechanisms | Refs. |
|---|---|---|---|
| | | **Spinal Cord Injury (SCI)** | |
| **DRUGS** | Valproic acid | M2 polarization<br>HDAC3, STAT1 ↓<br>TNF-α, IL-1β, IL-6, IFN-γ ↓ | [87] |
| | DL-3-*n*-butylphthalide (DL-NBP) | Motor function ↑, oedema ↓ | [90] |
| | Bradykinin B2 receptor antagonist-<br>HOE-140 | SC blood flow ↑; nNOS ↑<br>BSCB disruption ↓, oedema formation ↓ | [91] |
| | Protocatechuic acid (PCA) | Apoptotic cell death of neurons and oligodendrocytes ↓<br>Infiltration of neutrophils and macrophages ↓<br>MMP-9 ↓<br>TNFα, IL-1β, cyclooxygenase-2, inducible nitric oxide synthase and Chemokines ↓ | [83] |
| | Brilliant blue G (BBG) | P2X7, NLRP3, ASC, cleaved XIAP, caspase-1, caspase-11, IL-1ß, IL-18 ↑ | [92] |
| | Human immunoglobulin G (hIgG) | Antagonize neutrophil infiltration<br>Neutrophil chemo-attractants ↑ | [93,94] |
| | Haem oxygenase (HO)-1 | 4-Hydroxynonenal (4-HNE), malondialdehyde (MDA) ↑ | [95] |
| | NaHS<br>(H2S donor) | TJ proteins ↑<br>BSCB permeability ↓ | [96] |
| | Gallic acid (GA) | Jmjd3 ↓, MMP9 ↓<br>Neutrophil and macrophage infiltration ↓ | [97] |
| | Dl-3-*n*-butylphthalide (NBP) | ER stress ↓<br>Occludin, p120-Catenin, β-Catenin, claudin-5 ↑ | [98] |
| | Lithium chloride (LiCl) | Occludin, claudin-5 ↑<br>ER stress ↓<br>LC3-II, ATG-5 ↑ p62 ↓ | [99,100] |
| | Lycopene | Water content ↓<br>TNF-α and NF-kB ↓<br>ZO-1, claudin-5 ↑ | [101] |
| | Phenylbutyrate (PBA) | p120, β-catenin, occludin, claudin5 ↑<br>ER stress ↓<br>BSCB permeability ↓ | [102,103] |
| | Bromodomain and extra-terminal<br>domain (BET) proteins | Pro-inflammatory mediators ↓<br>Anti-inflammatory cytokines ↑<br>Reactivity of microglia/macrophages ↓<br>Neuroprotection and functional recovery ↑ | [104] |
| | Folic acid (FA) | MMP2 ↓ | [105] |
| | Flufenamic acid | TrpM4 ↓<br>MMP2 ↓, MMP9 ↓ | [84] |
| | Valproic acid | Microglia polarization; ↓TNF-α, IL-1β, IL-6, INF-γ | [87] |
| | Oleanolic acid (OA) | Caspase-3 ↓<br>Pro-inflammatory response ↓<br>MAPKs, NF-κB ↓ | [88] |
| | Salvianolic acid (A and B) | ZO-1, occludin ↑<br>TNF-α and NF-κB ↓<br>miR-101/Cul3/Nrf2/HO-1 ↑ | [89,106] |
| | Fluoxetine | MMP2 ↓, MMP9 ↓<br>ZO-1, occludin ↑<br>Groα, MIP1α and 1β ↓<br>Infiltration of neutrophils and macrophage ↓ | [107,108] |
| | Simvastatin-ezetimibe | ICAM-1 ↓<br>Endothelial inflammatory response ↓<br>Wnt/β-catenin ↑ | [109,110] |
| | Retinoic acid (RA) | P120, β-catenin, occludin and claudin5 ↑<br>CHOP ↓, caspase12 ↓ | [111] |
| | Epidermal growth factor (EGF) | Bax ↓, Bcl-2 ↑<br>Superoxide dismutase (SOD) ↑ glutathione peroxidase (GPx) ↑ | [112] |

**Table 3.** *Cont.*

| | Intervention | Mechanisms | Refs. |
|---|---|---|---|
| | Basic fibroblast growth factor (bFGF) | MMP9 ↓<br>Caveolin-1, TJs, including occludin, claudin-5, p120-catenin and β-catenin ↑<br>PRDX1 ↑→ autophagy<br>Neuroprotection ↑, axonal regeneration | [113,114] |
| | Methylprednisolone (MP) and aminoguanidine (AG) | AQP4 ↓<br>iNOS ↓ | [115,116] |
| | Tetramethylpyrazine (TMP) | BSCB permeability ↓<br>IL-1β, TNFα, IL-18, TUNEL-positive cells, caspase 3/9 ↓ | [117] |
| | Curcumin | TNF-α and NF-κB ↓<br>ZO-1, occludin ↑ | [118] |
| | Eugenol | NF-κB, p38 MAPK ↓<br>Inflammation, oxidative stress ↓ | [119] |
| | Mithramycin A (MA) Ghrelin 17β-estradiol (E2) | MMP9 ↓ SUR1/TRPM4 ↓<br>TJs ↑ | [81,120] |
| NANOPARTICLES | Carbon monoxide-releasing molecule-2 (CORM-2) | ZO-1, ZO-2, occludin and claudin-1 ↑ BSCB permeability ↓ | [121] |
| | CORM-3 | TJs ↑, MMP9 ↓ | [122] |
| | Bone mesenchymal stem cell-derived extracellular vesicles (BMSC-EV) | Brain cell death ↓<br>Neuronal survival ↑, motor function ↑<br>Pericyte migration ↓ | [123] |
| | Poly (D,L-lactide co-glycolide, PLGA)-based NPs | Localization at lesion site | [124–126] |
| BIOMATERIALS | Astragoloside IV Loaded Polycaprolactone Membrane | Caspase3 ↓, Bax/Bcl-2 ↓<br>Occludin, claudin5, ZO-1 ↑<br>MMP9 ↓, neutrophil infiltration ↓<br>BSCB permeability ↓ | [127] |
| | MSCs | BSCB leakage ↓<br>von Willebrand factor (vWF) ↑<br>Locomotor function ↑ | [128] |
| | HAMC/PLGA/FGF2 | FGF2 ↑ | [129] |
| miRNAs | miRNA-125a-5p | ZO-1, occludin, VE-cadherin ↑<br>BSCB permeability ↓ | [130] |
| | miR-429 | ZO-1, occludin and claudin-5 ↑<br>Krüppel-like factor 6 (KLF6) ↓ | [131,132] |
| | Ad-GFP-HO-1C[INCREMENT]23 | Hindlimb function ↑<br>TJs ↑ | [133] |

**Amyotrophic Lateral Sclerosis (ALS)**

| | Intervention | Mechanisms | Refs. |
|---|---|---|---|
| DRUGS | APC (Activated protein C) | IgG and iron deposition ↓<br>ZO-1, occludin ↑ | [134,135] |
| | AMD3100 | CXCR4/CXCL12 ↓<br>Microglial pathology ↑<br>Proinflammatory cytokines ↓ | [136] |
| BIOMATERIALS | Unmodified human bone marrow CD34+ (hBM34+) stem cells | EB extravasation ↓<br>Restored capillary ultrastructure<br>Engrafted widely into capillaries of the gray/white matter SC and brain Motor cortex/brainstem<br>structural and functional repair of BSCB impairment | [137–139] |
| | Human bone marrow-derived endothelial progenitor cells (hBMEPCs) | VEGF-A and angiogenin-1 ↑<br>EC phenotype ↑<br>ZO-1, occludin ↑ | [140] |
| | Mesenchymal stem cells (MSCs) | Motor neuron loss ↓<br>Locomotor activity ↑<br>Neurturin ↑ | [141] |

**Table 3.** *Cont.*

| | | | |
|---|---|---|---|
| **Spinal Cord Ischemia** | | | |
| DRUGS | Propofol | BSCB permeability ↓, MMP9 ↓, NF-kB ↓ Occludin ↑, claudin-5 ↑ | [142] |
| | Dexmedetomidine (Dex) | HMGB1-TLR4-NF-κB signalling pathway ↓ MMP9 ↓, angiopoietin-1 (Ang1) and Tie2 ↑ | [143,144] |
| | Remote ischemic preconditioning (RIPC) | Cannabinoid-1,2 receptors ↑ BSCB integrity ↑ ZO-1 ↑ MMP9 ↓, TNF-α ↓ | [145,146] |
| | Sevoflurane | MMP9 ↓ CXCL10, CCL2 ↓ IL-1β ↓ | [147] |
| BIOMATERIALS | BM-MSCs | EB extravasation ↓ MMP9 ↓, TNF-α ↓ | [148,149] |
| miRNA | miR-128-3p | Specificity protein 1 (SP1) ↓ Neuroinflammation ↓ | [150] |
| | miR-320a | AQP1 ↓ | [151] |
| | miR-27a | TICAM-2 ↓→NF-κB ↓ | [152] |
| **Multiple Sclerosis** | | | |
| DRUGS | Tetramethylpyrazine (TMP) | TNF-α, IL-1β ↓ IL-4, IL-10 ↑; TJs ↑ STAT3/SOCS3 ↑→NF-κB ↓→M2 polarization | [117] |
| | Calcitriol (vitamin D analog) | NLRP3, caspase-1, (IL)-1β, CX3CR1, CCL17, RORc, Tbx21 ↓ MHCII ↓ ZO-1 ↑ | [153] |
| | ADAMTS13 | VWF ↓ Demyelination ↓ T lymphocyte, neutrophil and monocyte infiltration ↓ | [154] |
| | Glyceryl tribenzoate (GTB) and Cinnamon | Perivascular cuffing ↓ Inflammation ↓ TGF-β, regulatory T cells (Tregs) in splenocytes ↑ | [155,156] |
| BIOMATERIALS | MSCs-IFN-β+minocycline | IFN-γ, TNF-α ↓ IL-4, IL-10 ↑ MMP9 ↓ | [157] |
| **Peripheral Nerve Injury (PNI)** | | | |
| | Salmon thrombin | TNF-α-induced endothelial permeability ↓ BSCB breakdown ↓ | [158] |

HDACs: Histone deacetylases; STATs: Signal transducers and activators of transcriptions; TNF: Tumour necrosis factor; IL: Interleukin; IFN: Interferon; nNOS: nitric oxide synthase; MMPs: Matrix metalloproteinases; NLR: NOD-like receptor; ASC: apoptosis-associated speck-like protein containing CARD; JMJD3:Jumonji domain-containing protein D3; ER: Endoplasmic reticulum; LC3: Microtubule-associated protein light chain 3; ATG5: Autophagy related 5; ZO-1: Zonula occludens-1; TrpM4: transient receptor potential cation channel subfamily M member 4; NF-κB: Nuclear factor-κB; MAPK4: mitogen-activated protein kinase; HO-1: heme oxygenase-1; Cul3:Cullin 3; Nrf2: Nuclear factor-erythroid factor 2-related factor 2; GROα:Growth-regulated oncogene α; MIP1α:macrophage inflammatory proteins; ICAM-1: Intercellular Adhesion Molecule 1; CHOP: CCAAT-enhancer-binding protein homologous protein; PRDX1:Peroxiredoxin 1; AQP4: Aquaporin 4; IgG: Immunoglobulin; CXCR: C-X-C chemokine receptor type; VEGF: Vascular endothelial growth factor; SOCS3:Suppressor Of Cytokine Signalling 3; RORC:RAR Related Orphan Receptor C; MHC: major histocompatibility complex.

Delivery systems for drugs across the BSCB have evolved over the past few years to enhance the transfer of therapeutics specifically to the sites of injury. Nanoparticles are well-established drug delivery systems owing to their nano-size, drug encapsulation capa-

bility, sustained drug release and biocompatibility [159] and have been used extensively to transport drugs across BBB (see Teleanu et al. [160]). To this end, various phase I/II clinical trials have been initiated using diverse nanoparticle–drug conjugates and are underway to target brain tumours through the BBB [161]. In line with the potential of nanoparticles to transverse BBB, some studies have also shown similar results transport across the BSCB. Nanoparticles originating from both biological (exosomes) and synthetic (lipids) sources have improved motor functions and restored TJs to attenuate BSCB leakage [121,162,163]. Other types of nanoparticles ranging from metals (e.g., iron oxide, gadolinium, cobalt, gold) and polymers to lipids have shown great potential both as tracers and drug-delivery systems in SCI (see Zuidema et al. [164]). Delivery of anti-inflammatory drugs such as methylprednisolone via nanoparticles significantly reduced lesion volume and improved behavioural outcomes when compared to delivery not guided by nanoparticles [165]. Another drug, flavopiridol, delivered using Poly (lactic-co-glycolic acid), PLGA-nanoparticles also significantly reduced inflammatory factors (e.g., TNF-$\alpha$, IL-1$\beta$, IL-6) and enhanced neuronal regeneration in a SCI mouse model [166]. Additionally, PLGA-nanoparticles encapsulated in hydrogel scaffolds (hyaluronan and methylcellulose) are non-toxic and can be used for sustained drug delivery to the injury site [167]. However, with limitations of nanomaterials such as toxicity and ambiguity in systemic clearance, it is prudent that other biological nanoparticles such as exosomes are also assessed as drug delivery carriers. Additionally, grey matter has been observed to be the most affected tissue in SCI and therefore it is wise to design modalities that target smaller, deeper vascular repair in the spinal cord [168]. However, these studies are limited to investigations over short time-intervals (30min-5hours) since advanced imaging techniques such as DCE-MRI have been used to show that disruption of the BSCB occurs for as long as 56 days post SCI [169].

*4.2. ALS*

ALS is a fatal neurodegenerative disease characterised by motor neuron degeneration in the brain and spinal cord, causing progressive paralysis and premature death typically within 3–5 years from diagnosis [170]. The majority of ALS cases (90–95%) are sporadic (sALS) and arise from an unknown origin, while the remaining 5–10% are genetically linked or familial ALS (fALS). Of fALS cases, 20% are associated with a missense mutation in the Cu/Zn superoxide dismutase 1 (*SOD1*) gene [171,172]. Other associated gene mutations include chromosome 9 open reading frame 72 (*C9orf72*), fused in sarcoma (*FUS*), angiogenin (*ANG*) and TAR DNA binding protein 43 (*TDP-43*) [173]. Despite genetic variance, both sALS and fALS share most clinical and pathological presentations. Early studies of ALS patients indicated possible BSCB impairment through elevated levels of cerebrospinal fluid (CSF) proteins [174–177] and greater CSF:serum albumin ratios [176,178–180]. Recent proteomic analyses support these results by revealing significant differences in CSF protein expression in ALS patients compared to healthy controls [181], reinforcing the importance of BSCB leakage in ALS. Multiple studies show associations between compromised BSCB structural components (astrocytes, neurons, ECs, pericytes) and ALS (Table 2). However, there is uncertainty as to whether such changes are causative or a consequence of disease progression.

As BSCB dysfunction likely contributes to ALS pathogenesis, several studies have explored different therapeutic approaches to repair and maintain BSCB integrity. Zhong et al. [135] initially utilised activated protein C (APC), an intrinsic plasma protease with anticoagulant properties, to restor TJ (ZO-1 and occludin) expression along with no IgG leakage or microhaemorrhages in SOD1$^{G93A}$ mice. These results were supported by Winkler et al. [134] who used an APC mutant, 5A-APC, to significantly improve BSCB integrity and delay the onset of motor impairment in SOD1$^{G93A}$ mice. 5A-APC treatment similarly improved BSCB structure and function by restoring normal levels of TJ proteins including ZO-1, occludin and claudin-5 as well as eliminating IgG and free iron deposits secondary to microhaemorrhage [134]. Following this, multiple studies (Table 3) investigated intravenously transplanted unmodified human bone marrow derived CD34+ (hBM34+) stem cells and their capacity to restore BSCB integrity in ALS mice [137–139]. This significantly

reduced microhaemorrhages [137], astrogliosis, microgliosis, capillary permeability and re-established perivascular end-feed astrocytes, as well as maintained motor neuron survival and delayed disease progression [139]. Together, these studies demonstrating improved ALS disease outcomes mediated through BSCB restoration further support the central role BSCB dysfunction in ALS. Future studies should continue to explore therapeutic approaches that support and re-enforce key structures of the BSCB, restricting access to neurotoxic substances which facilitate the ALS disease process.

### 4.3. PNI

Disruption of BSCB has been thought to be one of the earliest events following PNI. Initial evidence by Echeverry et al. [64], showed induction of neuropathic pain was due to recruitment of spinal blood borne monocytes/macrophages as a result of BSCB leakage. This was evidenced by presence of variably sized tracers (Evans blue/sodium fluorescein) found until 4 weeks post injury. This impairment of BSCB resulted from the release of a chemokine, monocyte chemoattractant protein-1 (MCP-1), from damaged neurons which regulates permeability in a transient and restricted manner as well as acting as a trigger for microglial activation that initiates neuropathic pain [65]. This study also reported contrasting roles of anti-inflammatory cytokines like TGFβ-1 and IL-10, which had the ability to repair the BSCB leakage post PNI. To evaluate the time-course of BSCB permeability, Cahill et al. [33], used in vivo DCE-MRI technique to show late onset BSCB permeability, which lasted for only one day post-surgery. Further, to understand the impact of variable genetic make-up on BSCB permeability and pain hypersensitivity, they assessed PNI in 5 different mouse strains (B10, C57BL/6J, CD-1, A/J and BALB/c). Interestingly, differences in permeability of Evans blue were observed amongst the different strains wherein increased up-take was observed specifically in CD-1 and A/J mice post-PNI. However, no strain related genetic correlations were observed in PNI-induced tactile hypersensitivity. Although limited, these studies do clearly indicate BSCB leakage following PNI in mice. Anti-inflammatory cytokines TGF-β1 (rescues occludin and ZO-1) and IL-10 were able to shut down the openings of the BSCB following PNI [64].

### 4.4. IRI

Spinal cord ischemia reperfusion injury (IRI) or SI is an adverse repercussion of thoracic aortic surgery and is closely associated with BSCB dysfunction. To-date, only one study [182] has reported impairment of BSCB that is attributed to the bimodal phase of SI. This study was based on a previously established hypothesis showing TLR-4 as an important transmembrane protein to be associated with inflammation post-SI. To further elucidate the role of TLR-4 related pathways in BSCB leakage, the expression of effector molecules such as myeloid differentiation factor 88 (MyD88) and TIR domain-containing adaptor inducing IFN-β (TRIF) were also evaluated. This showed the dependence of TLR4/MyD88 microglia-dependent activation in the first phase of SI, whereas TLR4/TRIF activation was related to the late phase, with the involvement of both microglia and astrocytes. These results emphasize the need for developing drugs for variable phases of SI. These observations corroborate with extravasations of Evans blue dye that too showed a pattern of bimodal distribution in the early stage i.e., 6–18 h post-surgery and later stage 36–48 h, which was in concurred with clinical manifestations. This team was also the first to show the effectiveness of bone marrow stromal cells in reducing inflammation in the BSCB in a SI-rabbit model [148]. Here too, BSCB leakage was recorded using Evans blue extravasation at disease onset with decreases in the TJ protein occludin and increases in MMP-9 and TNF-α levels. However, on exogenous treatment with bone marrow stromal cells, increases in the levels of occludin and inhibition of MMP-9 and TNF-α were observed, indicating restoration of BSCB integrity. Such activities of stromal cells may be attributed to their inherent anti-inflammatory properties and reduce TNF-α expression as seen in lipopolysaccharide-induced in vitro/in vivo inflammation models [183].

### 4.5. MS

MS is an acquired autoimmune disorder that results from changes in motor function due to effects on the brain and spinal cord. This disease has been mostly studied using an experimental animal model, EAE, that represents the physiology of human inflammatory demyelinating diseases such as MS. The pathophysiology of MS involves multifocal demyelination and neuronal loss which is attributed to the influx of activated immune cells such as leukocytes, T cells and macrophages into the CNS due to the impairment of the BBB or BSCB [184]. Although, substantial evidence establishes BBB leakage as one of the major causes of MS, few studies have examined the effects of BSCB disruption. In this regard, claudin-11, a transmembrane protein of CNS barriers (BBB, BSCB, the arachnoid barrier (BAB), BCSF), was studied to decipher its involvement in membrane disruption. Experimentation showed that downregulation of claudin-11 in the brain and spinal cord capillaries of an MS patient and EAE mice suggesting its involvement in maintaining the integrity of the BBB and BSCB [70]. One study evaluated the utility of MRI to detect lesions and BSCB disruption using the EAE model and showed the greatest disruption, predominantly in the white matter, at the onset of the disease, which declined as the disease further progressed [185]. Using the same model, this mechanism was further validated by Aube et al. [71], wherein disintegration of BSCB was observed within a day of disease onset in concurrence with the GFP$^+$ myeloid cell infiltration into the CNS that lasted for 4 days. A major observation was that permeability of the BSCB to small tracer molecules was due to the recruitment of neutrophils in the lumbar spinal cord. The role of neutrophils in BSCB disruption was validated by depleting neutrophils in the EAE mouse model using Anti-Ly6G treatment, which showed delayed manifestation of clinical symptoms of EAE and also decreased disease severity. Furthermore, the depletion of neutrophils also reduced BSCB permeability as evidenced by reduced tracer extravasation in these mice. Activated microglia also mediates BSCB impairment eventually leading to MS.

As a therapeutic agent, tetramethylpyrazine (TMP) has been suggested as an inhibitor of glial activation by modulating microglia polarization from the M1 to M2 phenotype through activation of STAT3/SOCS3 and inhibition of NF-κB signalling pathways. Further, TMP also restores TJ proteins and decreases expression of pro-inflammatory cytokines whilst increasing the expression of ant-inflammatory cytokines [26]. In EAE models, polarization of neutrophils in the CNS has a negative impact on BCSB integrity as also observed in BBB [186]. Further, compounds such as insulin-like growth factor (IGF-1) and erythropoietin significantly improved BSCB permeability and reduced axonal damage [182,187] (Table 3). Natalizumab, a human monoclonal antibody blocks interactions between α4 (on leukocytes) and its ligand on the BBB significantly reducing the disease impact [184,188]. However, its impact on spinal cord associated-MS has not been studied. Trials in future may be designed to understand its effect on the BSCB.

### 4.6. SCM

SCMs, also called cavernous angiomas and cavernomas, are relatively rare intramedullary vascular lesions found in 1.86 per 100,000 population and represents 5–12% of all spinal vascular malformations, and 3–5% of spinal cord lesions [189–194]. Pathologically, SCMs are characterised by well-circumscribed vascular malformations, which often appear on the spinal cord surface [195]. Histologically, SCMs consist of sinusoidal vascular spaces lined by a single layer of endothelial cells that are surrounded by loose connective tissue stroma, predisposing to haemorrhage [196]. Lesions in the spinal cord are of clinical interest as they are often surgically inaccessible and may lead to severe complications and even death [197]. Hence, a novel pharmacological approach is urgently needed for SCM patients. No studies have investigated the BSCB in SCM. However, in cerebral cavernous malformation, another type of cavernous malformation, exhibits incompetent and absence of blood-brain barrier (BBB) [198] and disrupted endothelium [199]. This shows that further investigations of the BSCB in SCM are warranted.

### 4.7. DCM

DCM is a progressive non-traumatic spinal cord disorder which results from compression in the neck and is typically diagnosed late due to non-specific and common symptoms that overlap with other neurological disorders [200]. The study of BSCB impairment in DCM is in its infancy and to-date only one study has detected this disruption in human subjects. Blume et al. [29], in a recently conducted prospective non-randomized study (*n* = 28) showed increased permeability of the BSCB in DCM patients through AlbuminQ expression in the intrathecal space as detected by concentrations of IgQs in CSF following a lumbar puncture. Interestingly, as opposed to other diseased models such as SCI and PNI, DCM showed a longer duration of clinical symptoms with disruptions of BSCB (6 months compared to 14 days–4 weeks). Differences in proteomic expression have also been observed in the CSF of dogs suffering from cervical spondylomyelopathy (CSM). CSM-affected dogs had enhanced expression of proteins responsible for actin regulation (vitamin D-binding protein, gelsolin), white matter damage and myelin degeneration (creatine kinase B-type), osteoarthritic changes (angiotensinogen, $\alpha$-2-HS-glycoprotein) and osteoarthritis (SPARC, calsyntenin-1, complement C3) that are known to be associated with BSCB impairment. Treatment with corticosteroid decreased the levels of angiotensinogen, $\alpha$-2-HS-glycoprotein and gelsolin and increased the expression of proteins associated with neuroprotection (transthyretin isoform 2, apolipoprotein E, cystatin C-like) and had anti-apoptotic effects (clusterin) [201]. DCM being an isolated spinal cord disorder may likely have a unique role of the BSBC. The spinal cord in the neck is subject to excessive motion the impact of motility on the vessel basement membranes and associated structures remains to be explored. An evolving concept around the causation of DCM is biomechanical and a putative role for the BSBC should be considered in the altered complex biomechanics of the area [202].

### 4.8. Cancers

Spinal cord cancer is a rare malignancy of the CNS. The majority (80–85%) of the CNS tumours occur in the brain with only 10–15% in the spinal cord. To date, a growing tumour mass has not been reported to be associated with disruption of the BSCB. On the contrary, impacts on the BSCB have been observed following anti-tumour treatment such as radiation therapy. Radiation injury occurs in the BBB where morphological changes in endothelial cells and TJ proteins has been observed both at early and late stages of radiation exposure [114,203,204]. Enhanced expression of the adhesion molecule, intercellular adhesion molecule-1 (ICAM-1) is implicated in the disruption of BBB integrity on irradiation [205]. ICAM-1, an important component of the TJs, has been observed in various injuries corresponding with BBB disruption and is inversely associated with barrier integrity [206,207]. To assess the effects of irradiation, Nordal et al. [208], examined the expression of ICAM-1 on BSCB disruption in rat spinal cords. They reported BSCB leakage of serum albumin that corresponded with increased expression of ICAM-1 on endothelial and glial cells 24 h after radiation injury. Although increased ICAM-1 expression is not an immediate response to irradiation, it is a downstream of other target genes such as various growth factors, cytokines and transcription factors which may be induced by radiation exposure. Hypoxia induced up-regulation of vascular endothelial growth factor in astrocytes has been observed post CNS radiation injury along with BSCB disruption as indicated by albumin extravasation in rat spinal cords [209,210].

## 5. Risk Factors

### 5.1. Aging

A recent finding by Piekarz et al. [211] shows significant loss (up to 41%) of alpha motor neurons ($\alpha$-MNs) with increasing age in mice. In addition, loss of myelin, compromised neuronal viability, increases in the age-related inflammation marker soluble ICAM-1 and the apoptotic marker caspase-3 are all indicative of the degenerative impact of aging on the

spinal cord. Furthermore, the authors observed greater permeability of BSCB to MMP-12 in aged mice which could be a contributing factor to increased apoptosis of $\alpha$-MNs.

### 5.2. Obesity and Metabolic Syndrome

Though the impact of obesity and metabolic syndromes such as diabetes have not been studied on the BSCB, evidence from the BBB suggest that related inflammation can cause impairment. Alterations in BBB transporters such as P-glycoprotein, low density lipoprotein receptor-related protein 1 and insulin transporters along with up-regulation of MMPs affect TJ proteins thereby causing BBB permeability [212]. However, in the BSCB, the little evidence that exists suggests that spinal cord dysfunction such as in SCI may significantly increase the risk of type 2 diabetes [213]. Further studies are warranted to establish a cause–effect relationship between metabolic disorders and the BSCB.

### 5.3. Lifestyle

Alcohol consumption has detrimental effect on BBB endothelial cells (in vitro) leading to gap formation between TJs along with GRP78 chaperone upregulation and increased ROS production [214]. In addition, evidence related to decreased TJ proteins (ZO-1, VE-cadherin, occludin), low-density lipoprotein receptor-related protein-1, receptor for advanced glycation end products, major facilitator superfamily domain-containing protein-2a and AQP4 with increased ROS production in mouse models (APPswe/PS1De9 mice) have demonstrated increased BBB permeability [215]. This detrimental effect of alcohol can be extrapolated to BSCB given presence of low TJs in its structural composition.

### 5.4. Infection and Auto-Immunity

Bacterial infections leading to meningitis influence BBB permeability by allowing flow of molecules such as albumin, nucleotide-binding oligomerization domain 2 (NOD2) and inflammatory factors [216,217]. *Borrelia burgdorferi*, the causative bacteria of Lyme disease also impacts cell-to-cell junctions including TJs and adherens junctions thereby influencing BCSF integrity [218]. More recently, it was observed that the vascular barrier in the choroid plexus (ChP) shuts down in response to inflammatory bowel disease via up-regulation of the Wnt/$\beta$-catenin pathway. This closure has been associated with mental deficits in models of genetically driven closure of ChP endothelial cells, highlighting the importance of peripheral interactions with the BCSF [219,220]. Furthermore, viruses such as human immunodeficiency virus type 1 (HIV-1) can target and alter the morphology and function of pericytes that constitute the BBB [221]. More recently, SARS-CoV-2 spike protein reportedly affected endothelial cells in the BBB via initiation of pro-inflammatory responses ultimately leading to leakage of pathogens, immune cells and cytokines into CSF and the brain in COVID-19 patients [222–225]. Other viruses such as Zika virus have shown evidence of penetration into brain parenchyma via alterations of TJ protein expression and disruption of BBB permeability [226,227]. Though, associations of bacterial and viral infections with BSCB integrity have not been reported, low abundance of TJ proteins in the BSCB could further aggravate its integrity and expedite damage to CNS.

### 5.5. Environment

Exposure to traffic related pollution has detrimental effects on the BBB. Suwannasual et al. showed that exposure to a mixture of gasoline and diesel vehicle engine emissions (MVE) led to alterations in BBB integrity (reduced TJs, upregulated IL-6, TGF-$\beta$) via Angiotensin-AT1 signalling and inflammation [228]. Similar effects on TJs were observed with environmental pollutant exposures such as perfluorooctane sulfonate [229] and pollution derived $Fe_3O_4$ nanoparticles [230]. Severe BBB dysfunction in children has been observed on exposure to dioxins such as 2,3,7,8-tetrachlorodibenzo-*p*-dioxin (TCDD) [231] and fine particulate matter (PM2.5) [232].

## 6. Conclusions

The terms BBB and BSCB have been often used interchangeably to define the blood-CNS barrier. However, evolving studies, show that BBB and BSCB are "two-sides of the same coin". Due to anatomical arrangements, the BSCB is often regarded as a mere extension of the BBB with similar structural components. Yet, lower numbers of pericytes and TJs in the BSCB make it more permeable than the BBB [20,21]. This phenomenon has been found in various spinal-cord related diseases such as SCI, PNI and SI along with neurological disorders such as ALS, MS and DCM. Such studies highlight that BSCB breach is as crucial a contributing factor to disease prognosis as is BBB. However, when compared to the BBB, BSCB investigations are limited and still lack evidence linking modulation of specific pathways or molecular/proteomic components.

MMP3/9/12-induced leakage is a crucial event in SCI, yet investigations pertaining to modulated pathways are lacking. In BBB models, elevated levels of MMP-9 break down TJ proteins and collagenase IV via the sonic hedgehog pathway (HH) transcription factor GLI-1 [233]. Furthermore, studies have also shown that the enzymatic activities of MMP9 are regulated by tissue inhibitors of metalloproteinase-1 (TIMP1) [234] and that the MMP9:TIMP1 ratio is a determinant of disease prognosis [235]. Other MMP-independent functions such as maintenance of barrier integrity is also maintained by TIMP1, which plays a key role in disease prognosis by regulating BBB permeability [236]. Considering that the BSCB is a functional equivalent of BBB, the postulated pathways such as HH and its downstream, targets such as TIMP1 and its effector molecules could be assessed to induce BSCB repair. Another interesting emerging aspect is the association of physical exercise with BBB/BSCB permeability. Strength and endurance training in an EAE model preserved TJ proteins in spinal cord tissue and also restricted the entry of autoreactive T cells into the CNS thereby maintaining BBB integrity [237]. Additionally, this study also showed inhibition of pro-inflammatory cytokines such as IFN-γ, IL-17 and IL-1β in the spinal cord strengthening the importance of physical exercise. Despite encouraging results, a few studies have also highlighted the negative outcomes of long-term exercise on BBB permeability [237,238]. Therefore, this aspect requires further clarification especially related to its influence on the BSCB interface and variable factors such as form and duration of exercise.

To further investigations and test potential therapeutic drugs on the BSCB, developing in vitro models mimicking this interface is crucial. In this regard, organoids with their 3-dimensional spatial arrangement can overcome the limitations of adherent 2-dimensional cell cultures and can be used to accurately predict the permeability of molecules/nanoparticles. Recently, organoids depicting the neurovascular BBB have been developed by Kumarasamy et al. [239] that express BBB-related genes, TJs, enzymes, proteins and structural components, and were observed to be more efficient in replicating cell–cell communication compared to 2D monocultures. Such findings open new avenues of study by enabling high-throughput screening of novel drugs and nanoparticles. Furthermore, bio-nanoparticles such as exosomes have shown great potential to transport drugs across the BBB [240]. Testing such drug-delivery vehicles to transport compounds across the BSCB are yet to be realised and should be tested.

## 7. Future Directions

The BSCB has gained substantial attention as an important component of the blood-CNS interface. With advancing technology, it is now important to understand the effect/causal relationship between BSCB damage and spinal disorders. To achieve a better understanding of BSCB pathophysiology, research on establishing iPSC-derived BSCB organoids would pave the way for small molecule drug screening and also realise the utility of repurposed drugs. A better understanding of the BSCB would also open new avenues for non-invasive therapeutics via delivery of nanoparticles to the spinal cord and also the brain. Evidence also suggests that lifestyle choices can metabolically regulate BSCB and more research should be focussed on this direction.

**Author Contributions:** Conceptualization, A.D. and N.C.; data curation, N.C., S.M. and A.D.; writing—original draft preparation, N.C.; writing—review and editing, N.C., S.M., J.P.C., P.M.H., A.D.D. and A.D.; supervision, A.D.D. and A.D. All authors have read and agreed to the published version of the manuscript.

**Funding:** P.M.H. is funded by a Fellowship and grants from the National Health and Medical Research Council (NHMRC) of Australia (1175134) and by UTS. A.D.D. receives unrestricted research grants from Nuvasive Australia & Baxter Inc, Education support from Globus Medical (PA). A.D.D. receives payments from Cartago Biotech, educational consultant payments from 3M & Nuvasive, research service payment from Kunovus Technologies & Merunova. This research received no external funding.

**Institutional Review Board Statement:** Not applicable.

**Informed Consent Statement:** Not applicable.

**Data Availability Statement:** Not applicable.

**Conflicts of Interest:** The authors declare no conflict of interest.

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
