# Peer review of "Blood-Spinal Cord Barrier: Its Role in Spinal Disorders and Emerging Therapeutic Strategies"

_neurosci, doi:10.3390/neurosci3010001_

Round 1
Reviewer 1 Report
This is a review of good quality which analyses the dysfunction of the blood-spinal cord barrier in several neurological disorders. The literature search was well conducted and the citations are appropriate. The themes are presented in a clear and consequent manner. I have no substantial changes to suggest, but rather minor stylistic points (1) wrong transcription of Greek letters; 2) line 237: "fuses in sarcoma" instead of "fused in sarcoma").
Author Response
1) This is a review of good quality which analyses the dysfunction of the blood-spinal cord barrier in several neurological disorders. The literature search was well conducted and the citations are appropriate. The themes are presented in a clear and consequent manner. I have no substantial changes to suggest, but rather minor stylistic points (1) wrong transcription of Greek letters.
Thank you for your valuable suggestions. We have rectified the Greek letters throughout the Manuscript.
2) line 237: "fuses in sarcoma" instead of "fused in sarcoma").
Done.
Reviewer 2 Report
In the current manuscript, Chopra and colleagues review the role of the blood-spinal cord barrier in brain disorders with spine-related pathology spine disorders and spine injuries. Here, the author’s present a quite exhaustive, well-written, and very timely summary of existing literature. From my side there are only a few minor comments that the authors may like to discuss or include in their manuscript.
- I know is it a matter of pure formalities but although CP is sometimes used as acronym for the choroid plexus, CP in a more neurological context is the standard acronym for Cerebral Palsy. Therefore, I would suggest the authors to use the also common ‘ChP’ acronym.
- Authors claim that the BCSFB has not studied in more detail. However, very recent studies have even provide a genetic evaluation of endothelial cells that seems to correlate with mental deficits in mouse models (Carloni et al., 2021). This work, in fact, highlight the importance of studying peripheral interactions with the BCSFB. This is a link that was suggested long ago (Marques et al., 2007; Marques et al., 2009), but currently be tested with more modern techniques.
- Previous findings showing immune recruitment to spinal cord lesions through or from the remotely located ChP (Kunis et al., 2013; Shechter et al., 2013), suggest that there is an interaction between the BCSB and the BCSFB for general CNS immune surveillance and repair. But less to no information about this is on the manuscript.
- In a recent study was shown that in MS patients pharmacological modulation with natalizumab prevents a further increase of the ChP volumes (Fleischer et al., 2021). Suggestion that drugs such as Natalizumab excerpt an effect over the BCSFB. Given that MS patients also largely show lesions at the spinal cord level, it may be suggested that Natalizumab plays also a role in modulating the BSCB.
- In MS patients, it appears a common finding that ChP is larger than in healthy control participants, and that this enlargements is also greater in patients with high lesion loads or active lesions at gadolinium enhanced MRI (Ricigliano et al., 2021; Fleischer et al., 2021). Moreover, ChP enlargement associates with higher uptake of translocator protein fluorine 18-DPA-714 PET, as marker of neuroinflammation (Ricigliano et al., 2021). Do the authors have information about such possibilities for the BCSB?
- Line 342, Repeated definition of “blood-cerebrospinal fluid barrier (BCSF)”
References
Carloni, S., Bertocchi, A., Mancinelli, S., Bellini, M., Erreni, M., Borreca, A., Braga, D., Giugliano, S., Mozzarelli, A. M., Manganaro, D., Fernandez Perez, D., Colombo, F., Di Sabatino, A., Pasini, D., Penna, G., Matteoli, M., Lodato, S., & Rescigno, M. (2021). Identification of a choroid plexus vascular barrier closing during intestinal inflammation. Science (New York, N.Y.), 374(6566), 439–448. https://doi.org/10.1126/science.abc6108
Fleischer, V., Gonzalez-Escamilla, G., Ciolac, D., Albrecht, P., Küry, P., Gruchot, J., Dietrich, M., Hecker, C., Müntefering, T., Bock, S., Oshaghi, M., Radetz, A., Cerina, M., Krämer, J., Wachsmuth, L., Faber, C., Lassmann, H., Ruck, T., Meuth, S. G., Muthuraman, M., … Groppa, S. (2021). Translational value of choroid plexus imaging for tracking neuroinflammation in mice and humans. Proceedings of the National Academy of Sciences of the United States of America, 118(36), e2025000118. https://doi.org/10.1073/pnas.2025000118
Kunis, G., Baruch, K., Rosenzweig, N., Kertser, A., Miller, O., Berkutzki, T., & Schwartz, M. (2013). IFN-γ-dependent activation of the brain's choroid plexus for CNS immune surveillance and repair. Brain : a journal of neurology, 136(Pt 11), 3427–3440. https://doi.org/10.1093/brain/awt259
Marques, F., Sousa, J. C., Coppola, G., Falcao, A. M., Rodrigues, A. J., Geschwind, D. H., Sousa, N., Correia-Neves, M., & Palha, J. A. (2009). Kinetic profile of the transcriptome changes induced in the choroid plexus by peripheral inflammation. Journal of cerebral blood flow and metabolism : official journal of the International Society of Cerebral Blood Flow and Metabolism, 29(5), 921–932. https://doi.org/10.1038/jcbfm.2009.15
Marques, F., Sousa, J. C., Correia-Neves, M., Oliveira, P., Sousa, N., & Palha, J. A. (2007). The choroid plexus response to peripheral inflammatory stimulus. Neuroscience, 144(2), 424–430. https://doi.org/10.1016/j.neuroscience.2006.09.029
Ricigliano, V., Morena, E., Colombi, A., Tonietto, M., Hamzaoui, M., Poirion, E., Bottlaender, M., Gervais, P., Louapre, C., Bodini, B., & Stankoff, B. (2021). Choroid Plexus Enlargement in Inflammatory Multiple Sclerosis: 3.0-T MRI and Translocator Protein PET Evaluation. Radiology, 301(1), 166–177. https://doi.org/10.1148/radiol.2021204426
Shechter, R., Miller, O., Yovel, G., Rosenzweig, N., London, A., Ruckh, J., Kim, K. W., Klein, E., Kalchenko, V., Bendel, P., Lira, S. A., Jung, S., & Schwartz, M. (2013). Recruitment of beneficial M2 macrophages to injured spinal cord is orchestrated by remote brain choroid plexus. Immunity, 38(3), 555–569. https://doi.org/10.1016/j.immuni.2013.02.012
Author Response
In the current manuscript, Chopra and colleagues review the role of the blood-spinal cord barrier in brain disorders with spine-related pathology spine disorders and spine injuries. Here, the author’s present a quite exhaustive, well-written, and very timely summary of existing literature. From my side there are only a few minor comments that the authors may like to discuss or include in their manuscript.
- I know is it a matter of pure formalities but although CP is sometimes used as acronym for the choroid plexus, CP in a more neurological context is the standard acronym for Cerebral Palsy. Therefore, I would suggest the authors to use the also common ‘ChP’ acronym.
Great suggestion. We have changed the acronym to “ÇhP” throughout.
- Authors claim that the BCSFB has not studied in more detail. However, very recent studies have even provide a genetic evaluation of endothelial cells that seems to correlate with mental deficits in mouse models (Carloni et al., 2021). This work, in fact, highlight the importance of studying peripheral interactions with the BCSFB. This is a link that was suggested long ago (Marques et al., 2007; Marques et al., 2009), but currently be tested with more modern techniques.
Thank you for raising this. We have inserted a discussion and these references concerning this in the manuscript (from Line 500).
- Previous findings showing immune recruitment to spinal cord lesions through or from the remotely located ChP (Kunis et al., 2013; Shechter et al., 2013), suggest that there is an interaction between the BCSB and the BCSFB for general CNS immune surveillance and repair. But less to no information about this is on the manuscript.
Thank you for raising this. We have inserted a discussion and these references concerning this in the manuscript (from line 229).
- In a recent study was shown that in MS patients pharmacological modulation with natalizumab prevents a further increase of the ChP volumes (Fleischer et al., 2021). Suggestion that drugs such as Natalizumab excerpt an effect over the BCSFB. Given that MS patients also largely show lesions at the spinal cord level, it may be suggested that Natalizumab plays also a role in modulating the BSCB.
Thank you for raising this. We have inserted a discussion and these references concerning this in the manuscript (from line 403).
- In MS patients, it appears a common finding that ChP is larger than in healthy control participants, and that this enlargements is also greater in patients with high lesion loads or active lesions at gadolinium enhanced MRI (Ricigliano et al., 2021; Fleischer et al., 2021). Moreover, ChP enlargement associates with higher uptake of translocator protein fluorine 18-DPA-714 PET, as marker of neuroinflammation (Ricigliano et al., 2021). Do the authors have information about such possibilities for the BCSB?
We do not have any information on this with regards to BSCB.
- Line 342, Repeated definition of “blood-cerebrospinal fluid barrier (BCSF)”
Corrected.
Reviewer 3 Report
This review will be of interest to those looking for an update on BSCB in the last decade, since the publication of Bartanusz et. al. (in accordance with the authors).
For those who have not read that review, it would be useful if the authors prepared tables describing key differences between the BBB and BSCB, especially if new differences have emerged in the last decade given the new assessment methodologies the authors describe. Given that the last segment of the review covers risk factors—mostly associated with BBB, not BSCB—a clearer understanding of how BBB studies potentially influences how we think about the BSCB is needed for these parts in particular.
Longer sections of text covering SCI and ALS should be split; as written, they cover a wide range of topics that are hard to follow.
Tables 1 and 2 do a lot of work to summarize mechanisms of BSCB impairment and interventions respectively, but much of the content is simply glossed over in the text. Pointedly, many of the references are cited only in the tables, and not in the main text. The authors should at least summarize the most important mechanisms that therapies seek to access. Additionally, listing the first author and year of publication in addition to the citation number would reduce some labor in finding appropriate citations.
Author Response
This review will be of interest to those looking for an update on BSCB in the last decade, since the publication of Bartanusz et. al. (in accordance with the authors).
1) For those who have not read that review, it would be useful if the authors prepared tables describing key differences between the BBB and BSCB, especially if new differences have emerged in the last decade given the new assessment methodologies the authors describe.
Response: Thank you for this suggestion. We have incorporated a new table (Table 1) depicting the differences.
2) Given that the last segment of the review covers risk factors—mostly associated with BBB, not BSCB—a clearer understanding of how BBB studies potentially influences how we think about the BSCB is needed for these parts in particular.
Response: We have rewritten this section to indicate why the BBB studies can be extended to BSCB.
3) Longer sections of text covering SCI and ALS should be split; as written, they cover a wide range of topics that are hard to follow.
Response: We have modified these sections to make them clearer.
4) Tables 1 and 2 do a lot of work to summarize mechanisms of BSCB impairment and interventions respectively, but much of the content is simply glossed over in the text. Pointedly, many of the references are cited only in the tables, and not in the main text. The authors should at least summarize the most important mechanisms that therapies seek to access. Additionally, listing the first author and year of publication in addition to the citation number would reduce some labor in finding appropriate citations.
Response: We have rewritten this section to include discussion of some of the references from Tables. We have followed the citation guidelines of the journal.